# Serum Level of Tumor-Overexpressed AGR2 Is Significantly Associated with Unfavorable Prognosis of Canine Malignant Mammary Tumors

**DOI:** 10.3390/ani11102923

**Published:** 2021-10-09

**Authors:** Stephen Hsien-Chi Yuan, Shih-Chieh Chang, Yenlin Huang, Hao-Ping Liu

**Affiliations:** 1Department of Veterinary Medicine, College of Veterinary Medicine, National Chung Hsing University, Taichung 40227, Taiwan; st851225@smail.nchu.edu.tw (S.H.-C.Y.); scchang@dragon.nchu.edu.tw (S.-C.C.); 2Department of Pathology, Chang Gung Memorial Hospital, Taoyuan 33378, Taiwan; louisyhuang@gmail.com

**Keywords:** canine, mammary tumors, AGR2, serum, tumor metastasis, prognosis

## Abstract

**Simple Summary:**

Canine malignant mammary tumor (MMT) is a prevalent malignancy in intact female dogs. A current lack of easily accessible tumor biomarkers hinders a timely assessment of the disease outcome. This study reveals that anterior gradient protein 2 (AGR2) is overexpressed in canine MMT tissues, and elevated levels of extracellular AGR2 in sera of MMT dogs are significantly associated with progression and remote metastasis of MMT and an unfavorable overall survival of the patients. Hence, serum eAGR2 level is significantly associated with an adverse outcome of MMT dogs and holds a predictive potential in MMT prognosis.

**Abstract:**

Canine malignant mammary tumors (MMTs) are prevalent malignancy in intact female dogs with a high incidence of metastasis and recurrence. A current lack of easily accessible tumor biomarkers hinders a timely assessment of the disease outcome. We previously identified anterior gradient protein 2 (AGR2) with higher protein abundance in canine MMT tissues compared with normal counterparts. AGR2 is an endoplasmic reticulum-resident protein disulfide isomerase involved in the regulation of protein processing and also exists extracellularly via secretion to exert pro-oncogenic functions. In the present study, we validated overexpression of AGR2 in canine MMT tissues from 45 dogs using immunohistochemistry and immunoblotting, and assessed serum AGR2 levels in 81 dogs with MMTs and 21 benign cases using a competitive enzyme-linked immunosorbent assay (ELISA). Our data revealed that serum eAGR2 levels are significantly correlated with MMT progression (*p* = 0.0007) and remote tumor metastasis (*p* = 0.002). Moreover, elevated levels of serum eAGR2 are associated with an unfavorable overall survival of MMT dogs in later stage (*p* = 0.0158). Area under the time-dependent ROC curve (AUC) of serum eAGR2 level as a prognostic indicator was 0.839. Collectively, this study uncovered that serum eAGR2 level is significantly associated with an adverse outcome of MMT dogs and holds a predictive potential in MMT prognosis.

## 1. Introduction

Canine mammary tumors (MTs) are one of the most prevalent neoplasms in intact female dogs, approximately 50% of which are malignant [1]. Most of the malignant MTs (MMTs) are of epithelial origin and classified into several subtypes. Prospective studies show that the MMT-associated death after 2 years of follow-up is 20–45% [2]. Despite an increasing number of studies shedding light on certain aspects of carcinogenesis of MMT, the molecular mechanisms underlying the pathogenesis of MMT still largely remains uncharacterized in contrast to human breast cancer. To identify new canine MMT-associated proteins that may contribute to the pathogenesis of this disease, we exploited quantitative proteomics analysis of canine MMT tissues and identified a list of proteins with higher abundance in canine MMT tissues compared with normal counterparts, including anterior gradient 2 (AGR2) [3].

AGR2 was first identified in *Xenopus laevis* as XAG-2, a secreted protein vital in developmental processes [4]. AGR2 belongs to the protein disulfide isomerase (PDI) superfamily that resides in the endoplasmic reticulum (ER) and mediates the formation of disulfide bonds, catalyzes the cysteine-based redox reactions, and modulates the quality control of proteins [5]. Proteins destined to the secretory pathway or to the cell surface are synthesized in the ER, where they undergo post-translational modification and proper folding executed by PDIs. For instance, production of the intestinal mucin MUC2 requires AGR2 via its thioredoxin-like domain to form intermolecular disulfide bonds with MUC2 [6]. Moreover, AGR2 binding to the epidermal growth factor receptor (EGFR) in the ER is crucial for EGFR delivery to the plasma membrane and consequential EGFR-transduced signaling [7]. Accordingly, misappropriation of PDIs such as AGR2 leads to accumulation of misfolded proteins in the ER and initiates ER stress that may drive disease development [8,9,10]. An increasing body of literature reveals the emerging biological roles of AGR2 in disease processes, e.g., inflammatory bowel diseases [11,12], cancer development and progression [5,13,14], and responses to chemotherapeutics [15,16,17,18]. AGR2 has been shown to be overexpressed in a variety of human adenocarcinomas wherein AGR2 promotes tumor growth and survival and is associated with adverse clinical outcomes [19,20,21,22,23].

Beyond aforementioned intracellular functions, AGR2 has also been reported to be localized on the cell surface of cancer cells and present in extracellular environments to exert distinct pro-oncogenic effects independently of its PDI activity [20,24]. Extracellular AGR2 (eAGR2) is found to act synergistically with insulin-like growth factor-1 (IGF-1) to induce cell proliferation, migration, and epithelial–mesenchymal transition in breast cancer cells [19]. eAGR2 also upregulates gene expression and activates the Wnt signaling pathway to promote cell migration of colorectal cancer (CRC) cells [25]. Of note, eAGR2 promotes the angiogenesis and invasion of vascular endothelial cells by directly interacting with vascular endothelial growth factor (VEGF) and fibroblast growth factor 2 (FGF2), thereby activating the VEGF receptor-mediated signaling [26,27]. It is noteworthy that eAGR2 is detectable in the body fluids of cancer patients and holds the potential for cancer diagnosis or prognosis [28,29,30]. Screening of eAGR2 hence provides a compelling avenue to predict tumor outcomes and guide therapeutic decisions. 

Despite the clinical relevance of eAGR2 in human neoplasms, the impact of eAGR2 on canine MMT has not yet been investigated. In this study, we evaluated the clinical significance of AGR2 in canine MMT and demonstrated the predictive value of serum eAGR2 in prognosis of this prevalent canine malignancy.

## 2. Materials and Methods

### 2.1. Tumor Tissue Specimens

Tissue specimens were collected from 45 privately owned female dogs with mammary tumors operated on simple or bilateral mastectomy at the Veterinary Medical Teaching Hospital (VMTH), National Chung Hsing University (NCHU), from 2015 to 2018. All methods were performed in accordance with relevant guidelines and regulations approved by the Institutional Animal Care and Use Committee (IACUC) of NCHU (IACUC Number: 107–004, 108–002, and 109–002). The patients were diagnosed with MTs by radiography and histopathological examination of involved tissues surgically removed. Tumor masses were cut into two parts, one of which was snap-frozen in liquid nitrogen immediately after resection and kept at −80°C until protein extraction for immunoblotting analysis; the other was fixed with formalin for histopathological confirmation. Classification, histopathological grade, and clinical stage of MMTs were determined on the basis of the modified WHO-TNM system [1]. 

For immunoblotting analysis, frozen MT tissues as well as paired normal mammary gland tissues from 11 MMT dogs were individually cubed into 5 mm in diameter and transferred to MagNA Lyser Green Beads (Roche, Basel, Switzerland) containing 200 µL of the lysis buffer (50 mM Tris-HCl (pH 8.0), 5 mM EDTA, 5 mM EGTA, 1% Triton X-100, 1 mM Na_3_VO_4_, 20 mM sodium pyrophosphate) and a protease inhibitor cocktail (VWR Life Science, Avantor, Radnor Township, PA, USA). Tissue grinding was performed immediately using MagNA Lyser (Rache) at 6500× *g* for 15 s. Resulting tissue lysate was transferred to a new microcentrifuge tube and centrifuged at 14,000 rpm at 4 °C for 10 min. Supernatants were collected and stored at −80 °C prior to further use. Protein concentration of the tissue lysate was measured using a BCA protein concentration assay kit (Pierce, Thermo Fisher Scientific, Waltham, MA, USA) following the manufacture’s protocol. For immunohistochemical analysis, tumor tissue slides of 34 MMT dogs were applied in this study.

### 2.2. Serum Collection

Pre-surgical serum samples were collected from 102 MT dogs diagnosed with MTs and operated on mastectomy at VMTH, NCHU from 2017 to 2020. Of these dogs, 17 cases overlapped with those for immunohistochemistry. All blood samples were placed in serum-separating tubes (SSTs) and left at RT for 30 min for clotting, and then were centrifuged at 2500× *g*, 4 °C, for 15 min. The collected sera were supplemented with a protease inhibitor cocktail (VWP Life Science, Avantor, Radnor Township, PA, USA), distributed into aliquots of 50 µL, and stored at −80 °C until utilization.

### 2.3. Immunohistochemical Analysis

Immunohistochemical analysis of AGR2 on canine MMT tissue slides was conducted using an automatic immunohistochemical staining device (BOND-MAX Automated Immunostainer; Vision BioSystems, Melbourne, Australia) with a mouse monoclonal antibody specific to AGR2 (Cat # MA5-16244, Invitrogen, Thermo Fisher Scientific; at 1:100 dilution) and BOND^TM^ Polymer Refine Detection Kit (Cat # DS9800, Leica, Wetzlar, Germany). AGR2 levels on individual tissue slides were assessed under 100× magnification and represented as the staining scores where the staining intensity of AGR2 (grading from 0 to 3) was multiplied by the proportion (%) of positively stained cells in the entire tissue section. On the basis of the scores, we classified AGR2 protein levels into three groups: low (<30), moderate (30–70), and high (≥70). IHC results were assessed and scored by a pathologist who was blinded to the pathological backgrounds of MMT patients.

### 2.4. Immunoblotting Analysis

Protein samples resolved in 1× sampling buffer (50 mM Tris-HCl, 1% ꞵ-mercaptoethanol, 2% SDS, 10% glycerol, 0.02% bromophenol blue, 50 mM EDTA (pH 6.8)) were denatured at 95 °C for 10 min, and then separated by sodium dodecyl sulfate polyacrylamide gel electrophoresis (SDS-PAGE) with 15% polyacrylamide gels. Protein samples were subsequently transferred to polyvinylidene difluoride (PVDF) membranes (GE Healthcare), which was blocked with HyBlock 1-min blocking buffer (GOAL Bio, Hycell International Co., Ltd., Taipei, Taiwan) at room temperature (RT) for 1 min, and incubated at 4 °C overnight with a mouse monoclonal antibody specific to AGR2 (Cat # MA5-16244, Invitrogen, Thermo Fisher Scientific; at 1:400 dilution), a mouse monoclonal antibody specific to glyceraldehyde 3-phosphate dehydrogenase (GAPDH; Cat # 60005-1-Ig, Proteintech; at 1:5000 dilution), or a mouse monoclonal antibody specific to a 6×His tag (Cat # 60004-1-Ig, Proteintech; at 1:5000 dilution). The membranes were then incubated with secondary horseradish peroxidase (HRP)-conjugated goat anti-mouse IgG (PerkinElmer, Waltham, MA, USA, at 1:10,000 dilution) at RT for 1 h and washed with 1× Tris-buffered saline containing 0.05% Tween-20 (TBS-T) between steps. Luminescence signals were developed with Western Lightning^®^ ECL-Pro (PerkinElmer, Waltham, MA, USA), and images were acquired using Hansor Luminescence Image System (Hansor Polymer Technology Corp., Taichung, Taiwan) with TSGel software (version 3.5). Quantification of protein bands was performed using ImageJ (version 1.50i).

### 2.5. Preparation of Recombinant Canine AGR2 Proteins

The pET-24a(+)-canine AGR2 was constructed by using a conventional PCR-based cloning method. First, total RNA was extracted from a canine MMT cell line DMGT by using NucleoZOL (MN, Düren, Germany) according to the manufacture’s protocol. mRNA was then reverse-transcribed into complementary DNA (cDNA) using the M-MuLV RTase (Protech, Taipei, Taiwan) at 42 °C for 1 h. The DNA fragment encoding amino acids 21–175 of canine AGR2 was generated by PCR using DMGT-derived cDNA as template together with a pair of primers: forward, 5′-GGTCATATGGATATCACAGTTAAATCAGG-3′; reverse, 5′-GAACTCGAGCAATTCAGTCTTTAGCAAC-3. The resulting DNA fragment was ligated into the pET-24a(+) vector via the *Nde* I and *Xho* I sites. The sequence accuracy of the construct was confirmed by an automatic DNA sequencing (Tri-I Biotech Inc., New Taipei City, Taiwan). Moreover, pET-24a(+)-canine AGR2 was introduced into *Escherichia coli* (*E. coli*) BL21 (DE3) for expression of recombinant canine AGR2 (rcAGR2) that was C-terminally tagged with 6× His. Expression of rcAGR2 was induced by 0.5 mM isopropyl thiogalactoside (IPTG), and rcAGR2 proteins were further purified by using the nickel–nitrilotriacetic acid (Ni-NTA) Sepharose^TM^ 6 Fast Flow resins (GE healthcare) according to the manufacturer’s instructions.

### 2.6. Establishment of a Competitive Enzyme-Linked Immunosorbent Assay (ELISA) for Serum eAGR2 Detection

rcAGR2 diluted in 1× phosphate-buffered saline (PBS) at a concentration of 250 ng/100 µL per well was coated onto a 96-well microplate (Iwaki®, Asahi Glass Co., Ltd., Tokyo, Japan) at 4 °C for 16 h, and excessive rcAGR2 was washed away with 1× PBS five times. The coated wells were blocked with BlockPRO^TM^-Protein-free solution (Visual Protein, Taipei, Taiwan; 100 µL/well) at RT for 2 h, and then washed with 1× PBS five times. A rabbit polyclonal antibody to AGR2 (Cat # PA5-34517, Invitrogen, Thermo Fisher Scientific) diluted at 1:6000 in 100 µL of BlockPRO^TM^ solution was incubated with canine serum samples (at 1:30 dilution) or rcAGR2 standards (8, 2, 0.5, 0.125, and 0.03125 µg/mL) diluted in 100 µL of BlockPRO^TM^ solution at RT for 1 h. Afterwards, the mixture was loaded into the coated microplate (100 µL per well) and incubated at RT for 1 h. After the microplate was washed five times with 1× PBS containing 0.05% Tween-20 (1× PBS-T), secondary HRP-conjugated goat-anti-rabbit IgG diluted at 1:10,000 in BlockPRO^TM^ solution was added into the microplate (150 µL per well), and the mixture was incubated at RT for 40 min. After washing steps, colorimetric development of the reaction was achieved by incubation with tetramethylbenzidine (TMB; Clinical Science Products, Mansfield, MA, USA; 100 μL/well) in the dark for 3 min, and then ceased by adding 2N sulfuric acid (50 μL/well). Optical density (OD) values of the reactions were measured using SPECTRO star Nano (BMG LABTECH, Ortenberg, Germany) at the wavelength of 450 nm. The OD_450_ values of samples in individual wells were corrected by subtracting the OD_540_ values of the same wells in accordance with the manufacturer’s instruction on the substrate TMB. The reaction in which the AGR2-specific antibody was omitted was used as a blank control. All samples were assayed in duplicate. The absorbance of samples was presented as (mean (OD_450_–OD_540_) of samples—mean (OD_450_–OD_540_) of the blank control).

Ten competitive ELISAs were conducted to analyze serum samples of 102 MT dogs enrolled in this study. In each assay, rcAGR2 standards and three fixed control canine serum samples were included, the latter of which were used for calibration of the data across 10 ELISAs. The eAGR2 concentration in serum samples was calculated according to the standard curve set with rcAGR2.

### 2.7. Precision Evaluation of the Established Competitive ELISA

The possible influence of canine serum in the competitive ELISA was evaluated by assaying rcAGR2, which was spiked at concentrations of 8, 2, 0.5, 0.125, and 0.03125 µg/mL in 30-fold diluted canine serum where eAGR2 was undetectable. The measured concentration of rcAGR2 spiked in canine serum was compared with its theoretical concentration to calculate the spike recovery rate as ((measured concentration of spiked rcAGR2/theoretical concentration of spiked rcAGR2) × 100%). Moreover, the intra-assay coefficient variant (CV) was determined by calculating the variation between measurements of duplicated samples in each ELISA. The inter-assay CV was determined by assaying three fixed serum samples across 10 ELISAs. The CV was calculated as ((the standard deviation (SD)/the mean value) × 100%). The detection sensitivity was determined by calculating the standard zero (0 µg/mL) of rcAGR2 across 10 ELISAs and set as (the mean + 2 × SD).

### 2.8. Cell Culture

Two canine MMT cell lines, DMGT and CF41.Mg, were used for verification of AGR2 expression. DMGT was previously established by culturing primary tumor cells derived from MMT tissue cubes ex vivo for more than 100 passages. CF41.Mg was purchased from American type culture collection (ATCC; CRL-6232^TM^). Both cell lines were maintained in Dulbecco’s high glucose modified Eagle’s medium (DMEM; Gibco, Thermo Fisher Scientific) supplemented with 10% fetal bovine serum (FBS; Gibco, Thermo Fisher Scientific) at 37 °C in an incubator supplied with 5% CO_2_.

### 2.9. Immunofluorescence Staining

DMGT and CF41.Mg cells were seeded onto coverslips placed in a 12-well plate and grown to 50–70% confluency. Cells were fixed with 4% paraformaldehyde containing 2% sucrose in 1× PBS at RT for 20 min and then permeabilized with 0.1% Triton X-100 for 3 min and blocked with BlockPRO^TM^ (Visual protein, Taipei, Taiwan) for 30 min. Cells were incubated with the AGR2 antibody (MA5-16244, Invitrogen, Thermo Fisher Scientific) at 1:100 dilution for 90 min at RT, followed by staining with secondary Alexa Fluor 488-conjugated goat-anti-mouse IgG (Molecular Probe, Thermo Fisher Scientific) at 1:200 dilution for 45 min. The nuclei were co-stained with 4′,6-diamidino-2-phenylindole (DAPI) at 0.1 µg/mL in 1× PBS. Coverslips were washed with 1× PBS between steps and finally mounted with Fluoro-Gel (Electron Microscopy Science, USA) on slides. Results were observed by using Leica DMI3000 Inverted Microscope (Leica, Wetzlar, Germany) equipped with a Zyla 5.5 mega pixel sCMOS camera (Andor Technology, Belfas, Ireland) and processed with MetaMorph^®^ NX Software (Molecular Devices, San Jose, CA, USA).

### 2.10. Collection of Conditioned Media and Cell Lysates of Canine MMT Cells

DMGT and CF41.Mg cells were grown in serum-free OPTI-MEM (Gibco, Thermo Fisher Scientific) for 24 h, and the media were collected and centrifuged at 5000 rpm for 5 min to remove detached cells. The media were subsequently concentrated and desalted with Vivaspin^®^ 6 sample concentrators (GE Healthcare). For cell lysate extraction, cells were washed with 1× PBS three times and lysed on ice with the lysis buffer (50 mM Tris-HCl (pH 8.0), 5 mM EGTA (Sigma-Aldrich, Merck, Burlington, MA, USA), 5 mM EDTA (Amresco-VWR, Radnor Township, PA, USA), 1% Triton X-100, 20 mM sodium pyrophosphate) freshly supplemented with a protease inhibitor cocktail (VWP Life Science, Avantor, Radnor Township, PA, USA) and phenyl methyl sulphonyl fluoride (PMSF; Sigma-Aldrich). Cell lysates were collected into a microcentrifuge tube and centrifuged at 12,000 rpm at 4 °C for 15 min, and the resulting supernatant was transferred to another microcentrifuge tube. Protein concentrations of the conditioned media and cell lysates were measured using a BCA protein concentration assay kit (Pierce, Thermo Fisher Scientific).

### 2.11. Follow-Up of the MT Patients

The follow-ups of 102 MT dogs after surgery were acquired by telephone interview every six months until the end of this study. Overall survival time (OST) was estimated as the survival period from surgery to death or until the end of the study (alive). The patients who died from a cause unrelated to tumor (e.g., accidence or heart diseases), died within 7 days after surgery, or were lost to follow-up were excluded. No patient was performed on euthanasia as a result of MMT.

### 2.12. Statistical Analysis

For the competitive ELISA, the standard curve of rcAGR2 was set using four parameters logistic regression where *R*^2^ > 0.99 was considered a fitted result. Comparison between two categories of data was conducted using the Mann–Whitney *U* test. Comparison among three categories of data was carried out using the Kruskal–Wallis *H* test. For overall survival analysis, the median of serum eAGR2 concentration (5.65 µg/mL) in 102 MT cases was used as the cutoff to dichotomize the analyzed cases into eAGR2^High^ (concentration ≥ 5.65 µg/mL) and eAGR2^Low^ (concentration < 5.65 µg/mL) groups. Comparison of overall survival (OS) between eAGR2^High^ and eAGR2^Low^ groups was conducted using the Kaplan–Meier method, and the significance was evaluated using the log rank test. All statistical analyses described above were performed by using GraphPad Prism V8.4 software (GraphPad Inc., San Diego, CA, USA). All *p*-values were two-tailed, and *p* < 0.05 was considered statistically significant. The performance of serum eAGR2 concentration for OS prediction was estimated by using the “survival ROC” package with a time-dependent ROC curve in R software as previously described [31].

## 3. Results

### 3.1. Elevated Protein Level of AGR2 in Canine Mammary Tumor Tissues

We first conducted immunohistochemistry (IHC) analysis to verify AGR2 protein expression in canine MMT tissues using tumor tissue slides of 34 MMT dogs. As shown in Figure 1A, AGR2 expression was detected in MMT tissues at a higher protein level compared with that in adjacent normal counterparts (Figure 1C, *p* < 0.001). No staining signal was detected on tumor tissues using a control IgG (Figure 1B). Additionally, aberrant localization of AGR2 at the tumor cell surface in MMT tissues was noted in some cases (Figure 1A). Overall, AGR2 protein was overexpressed in MMT tissues in contrast to adjacent normal mammary tissues (Figure 1D).

Furthermore, AGR2 overexpression in MMT tissues was also validated by immunoblotting analysis using MMT and paired normal mammary gland tissues of another 11 MMT dogs. Results revealed that AGR2 protein levels were significantly elevated in tumor tissues compared with paired normal mammary gland tissues collected from the same patient (Figure 2A,C), in consistence with the IHC data. Collectively, AGR2 was overexpressed in overall MMT tissues and even in BMT tissues compared with the normal counterparts (Figure 2B). These data indicated a significant association of AGR2 expression level with canine MMT.

### 3.2. Detection of AGR2 in Cell Lysates and Serum-Free Conditioned Media of Canine MMT Cells

To better examine the expression pattern of AGR2 in canine MMT cells, we performed immunofluorescence staining with two canine MMT cell lines. As shown in Figure 3A, AGR2 was primarily detected at the intracellular compartment. Immunoblotting analysis confirmed AGR2 protein expression in cell lysates of both cell lines (Figure 3B). To further verify if secretion of AGR2 takes place in canine MMT cell lines, we assessed extracellular AGR2 (eAGR2) in the serum-free culture media (conditioned media) of the two cell lines. The conditioned media were further concentrated and subjected to a competitive ELISA established for eAGR2 detection (Figure 4A). Amounts of eAGR2 were 20 ng and 40 ng, respectively, in 50 µg of total proteins in the concentrated media of two cell lines (Figure 3C). Results indicated that canine MMT cells secreted eAGR2 that was detectable in cell conditioned media.

### 3.3. Establishment of a Competitive ELISA for eAGR2 Detection

To detect eAGR2 in sera of MMT dogs, we established an in-house competitive ELISA, as illustrated in Figure 4A. We first constructed an expression vector for production of recombinant canine AGR2 (rcAGR2) to be coated onto the microplate. Identity of rcAGR2 was confirmed by immunoblotting using an AGR2-specific antibody (Figure 4B). To assess serum eAGR2 concentration with the ELISA, we set a standard curve by incubating the primary antibody specific to AGR2 (anti-AGR2) with rcAGR2 at concentrations of 8000, 2000, 500, 125, and 31.25 ng/mL. The standard curve built by utilizing 6000-fold diluted anti-AGR2 fitted a linear regression modeling the relationship between the OD values and the log-transformed rcAGR2 concentrations (Figure 4C). The standard curve set for each ELISA showed a consistency in performance across more than 10 experiments.

Next, we evaluated the precision and effectiveness of the established ELISA for eAGR2 detection. As shown in Figure 4D, the mean of the intra-assay coefficient variant (CV) of 10 assays was 2.861%, and the inter-assay CV was 3.586% across 10 assays. The sensitivity of the established ELISA was 0.0293 µg/mL. Furthermore, a spike recovery test, in which a canine serum without detectable eAGR2 was spiked with fourfold serially diluted rcAGR2 (from 8 to 0.03125 µg/mL), was conducted to assess the possible interference of canine serum components with the ELISA. Data revealed that the average recovery rate across different concentrations of spiked rcAGR2 was 93.8% (Figure 4E). Results demonstrated the precision of the established ELISA for eAGR2 detection.

### 3.4. Association of Serum eAGR2 Concentration with Stage and Metastasis of MMT

To evaluate the correlation between serum eAGR2 level and MMT progression, we conducted the competitive ELISA to analyze eAGR2 levels in pre-surgical serum samples from 102 MT dogs, including 81 MMT and 21 BMT cases, whose clinicopathological characteristics are summarized in Table 1. Of these dogs, 17 dogs overlapped with the cases for IHC analysis. In addition, eight dogs were administered with chemotherapy (doxorubicin, carboplatin, or cyclophosphamide) after surgery. Levels of eAGR2 in all analyzed serum samples lay within the detection range of the established ELISA. Results showed that serum eAGR2 concentrations in MMT dogs were significantly higher than that in the BMT cases (*p* < 0.05, Figure 5A). Of note, serum eAGR2 levels in dogs with metastatic MMT (stage IV/V) were significantly elevated compared with that in dogs with metastasis-free MMT (stage I–III, *p* < 0.01), as shown in Figure 5B. Moreover, the concentrations of serum eAGR2 were significantly increased in dogs with MMT exhibiting a remote metastasis (stage V), compared with that in dogs with MMT in stage III/IV (*p* < 0.01) or in stage I/II (*p* < 0.001), or those with BMT (*p* < 0.001, Figure 5C). Results collectively demonstrated that serum eAGR2 concentration was highly associated with canine MMT progression and distant metastasis.

Furthermore, we analyzed the correlation of serum eAGR2 concentration with clinicopathological factors of MMT cases, including age, neuter status, body weight, MMT subtype, clinical stage, and tumor metastasis. As shown in Table 2, serum eAGR2 concentration was significantly correlated with clinical stage and tumor metastasis of MMT (*p* = 0.0007 and 0.002, respectively), reinforcing the positive association of serum eAGR2 level with MMT progression. In contrast, there was no significant correlation of serum eAGR2 level with age, body weight, neuter status, or tumor subtype.

### 3.5. Significant Correlation of Serum eAGR2 Concentration with Unfavorable Overall Survival of MMT Dogs

We further investigated the prognostic value of serum eAGR2 in overall survival (OS) prediction of MMT cases. As shown in Figure 6A, of 102 MT dogs enrolled in this study, 24 dogs who were lost to follow-up or died from a cause unrelated to tumor and 5 dogs who died within seven days after surgery were excluded from the overall survival analysis (*n* = 73). The median of serum eAGR2 concentrations in 102 MT dogs was 5.65 µg/mL (range: 1.76–15.55 µg/mL). We utilized the median as the cutoff to dichotomize the enrolled cases into two groups, i.e., eAGR2^Low^ (<5.65 µg/mL) and eAGR2^High^ (≥5.65 µg/mL). Kaplan–Meier survival analysis of total 73 MT dogs revealed that eAGR2^High^ dogs had poorer OS compared with the eAGR2^Low^ cases (*p* = 0.0269, Figure 6B). Consistently, in 61 MMT dogs, the eAGR2^High^ group had a worse OS compared with the eAGR2^Low^ group (*p* = 0.0328, Figure 6C). To assess the influence of chemotherapy on OS of MMT patients, we compared OS between the dogs administered with chemotherapy after surgery (*n* = 8) and those without chemotherapy (*n* = 53). As shown in Appendix A, the chemotherapy group had a significantly worse OS compared with the chemotherapy-free group (*p* = 0.0002), suggesting that chemotherapy had no beneficial effect on OS of the patients. Instead, serum eAGR2 levels in the chemotherapy-administered group were much higher than in the chemotherapy-free group, reinforcing the correlation of serum eAGR2 level with adverse OS. 

To further evaluate the correlation between serum eAGR2 concentration and OS of MMT dogs in later clinical stages, we further applied the survival analysis to MMT patients in stage II to V. Results demonstrated that the eAGR2^High^ group had an unfavorable outcome compared with the eAGR2^Low^ group (*p* = 0.0158, Figure 6D). Furthermore, the predictive value of serum eAGR2 concentration in MMT prognosis was estimated by using a time-dependent receiver operating characteristic curve (ROC). As seen in Figure 6E, the area under the ROC curve (AUC) of serum eAGR2 concentration as a prognostic indicator was 0.839, which was superior to the AUC of age (0.429) and body weight (0.467) and comparable to that of stage (0.871). Collectively, these data showed that serum eAGR2 concentration was significantly associated with an adverse clinical outcome of MMT patients and holds a predictive potential in MMT prognosis.

## 4. Discussion

Although surgical resection is effective enough to confine early-stage and low-grade canine MMT [32], tumor metastasis and recurrence are the common causes of treatment failure in dogs with advanced-stage or high-grade MMT. Current prognostic evaluation of canine MMT mainly relies on tumor staging and histopathological grading [33,34], while the processes are relatively time-consuming and require expertise. In this regard, there is a need of easily accessible molecular biomarkers to ameliorate prognostic prediction of the disease outcome to benefit the patient welfare. 

We previously identified AGR2 with higher protein abundance in canine MMT tissues by conducting comparative quantitative proteomics analysis [3]. Canine AGR2 shares 94% homology with human AGR2 in the amino acid sequence of polypeptide where putative functional motifs are highly conserved. AGR2 has emerging pro-oncogenic roles in various types of human cancers [13,14,24,27]. In this study, we not only validated overexpression of AGR2 in canine MMT tissues compared with paired normal counterparts, but also assessed serum eAGR2 levels in MMT dogs using an in-house competitive ELISA. Our data revealed that serum eAGR2 levels were elevated in MMT dogs compared with BMT cases. Importantly, the serum eAGR2 level was significantly correlated with progression and remote metastasis of canine MMT. It is noteworthy that serum eAGR2 levels in dogs with advanced-stage MMTs were significantly associated with an adverse overall survival. Our findings are in line with the notions that elevated levels of eAGR2 in sera or plasma of human cancer patients are correlated with worse prognosis [29,35]. To our knowledge, our study is the first to reveal that serum eAGR2 has a potential to predict clinical outcome of canine MMT.

## 5. Conclusions

In summary, herein we demonstrated the predictive value of serum eAGR2 in prognosis of canine MMT. The present study also sheds new light on the potential involvement of AGR2 in the progression of canine MMT, which merits further efforts to better characterize the pro-metastatic functions of AGR2 in this prevalent canine malignancy.

## Figures and Tables

**Figure 1 animals-11-02923-f001:**
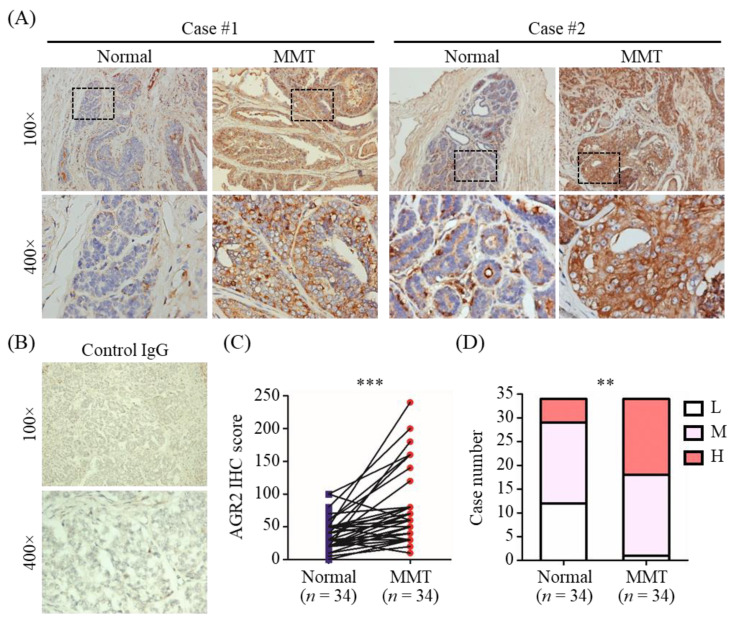
AGR2 overexpression in canine MMT tissues revealed by immunohistochemistry (IHC). AGR2 protein expression was analyzed with tissue slides of 34 canine MMT cases using an antibody specific to AGR2. (**A**) Micrographs of two representative cases. Upper micrographs were acquired at 100× magnification; bottom micrographs were the areas in the upper images acquired at 400× magnification. (**B**) MMT tissues stained with an isotype control IgG. (**C**) Pairwise comparison of IHC scores of AGR2 between MMT tissue and adjacent normal mammary gland tissue of the same patient. Each line represents an individual patient. Statistical significance was determined by two-tailed paired *t*-test. (**D**) Comparison of IHC scores of AGR2 between overall MMT tissues and adjacent normal mammary tissues. L, M, and H denote low, moderate, and high levels of AGR2, respectively. Statistical significance was determined by Fisher’s exact test. ** *p* < 0.01; *** *p* < 0.001.

**Figure 2 animals-11-02923-f002:**
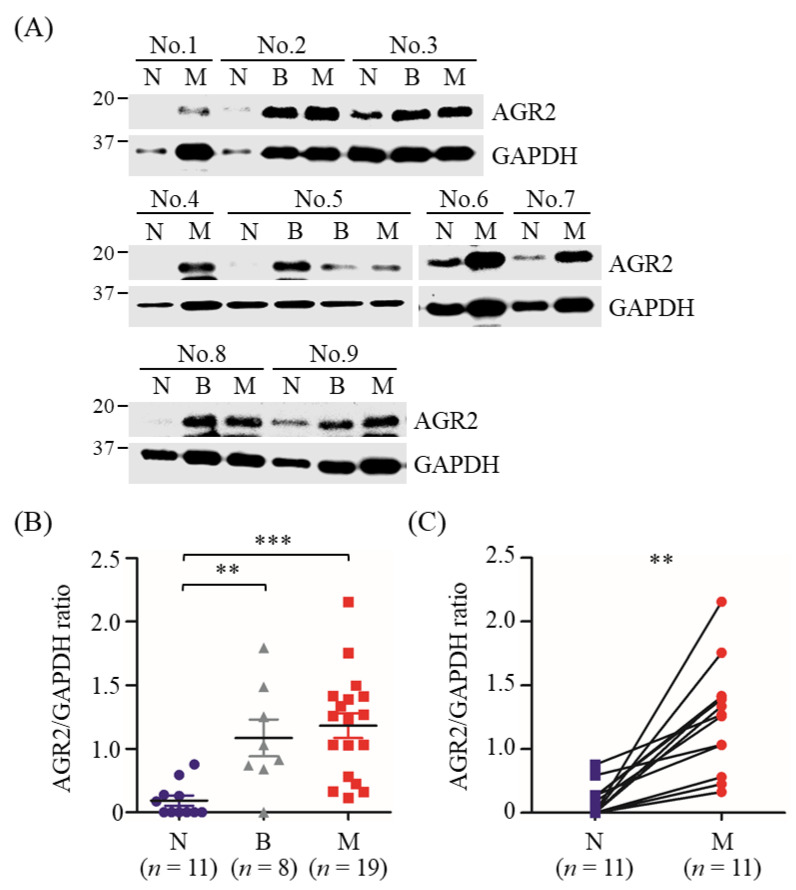
Elevated protein levels of AGR2 in canine MMT tissues compared with paired normal mammary tissues. Tissue homogenates (20 µg proteins each sample) of MMT (denoted M), paired with benign mammary tumor (denoted B) and normal mammary gland tissues (denoted N) collected from the same patient, were analyzed by immunoblotting using an antibody specific to AGR2. GAPDH was used as an internal control. Representative results of nine cases (No. 1 to No. 9) were shown in (**A**). AGR2 protein levels in individual samples were determined by calculating the AGR2-to-GAPDH ratio. (**B**) Comparison of AGR2 protein levels between tissue types. Statistical significance was determined by Mann–Whitney *U* test. (**C**) Pairwise comparison of AGR2 protein levels between MMT and paired normal mammary gland tissues of the same patient. Lines represent individual patients. Statistical significance was determined by two-tailed paired *t*-test. ** *p* < 0.01; *** *p* < 0.001.

**Figure 3 animals-11-02923-f003:**
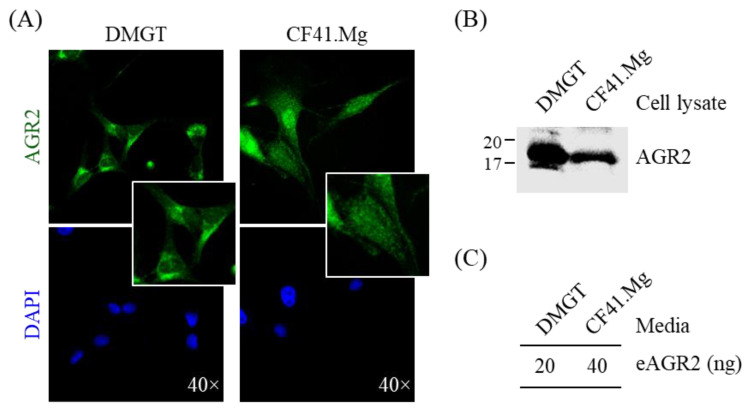
Detection of AGR2 expression in canine MMT cells and conditioned media. (**A**) Intracellular localization of AGR2 in canine MMT cell lines, DMGT, and CF41.Mg. AGR2 expression was verified by immunofluorescence staining using an AGR2-specific antibody. DAPI staining indicated the nuclei. Magnification, ×40. (**B**) Detection of AGR2 in cell lysates of canine MMT cell lines. Cell lysates (30 µg proteins per sample) were analyzed by immunoblotting with an AGR2-specific antibody. (**C**) Detection of extracellular AGR2 (eAGR2) in serum-free conditioned media of canine MMT cells. Cells were grown in serum-free media for one day, and the conditioned media were harvested and subsequently concentrated. The resulting conditioned media (50 µg proteins per sample) were analyzed using a competitive ELISA established for eAGR2 detection.

**Figure 4 animals-11-02923-f004:**
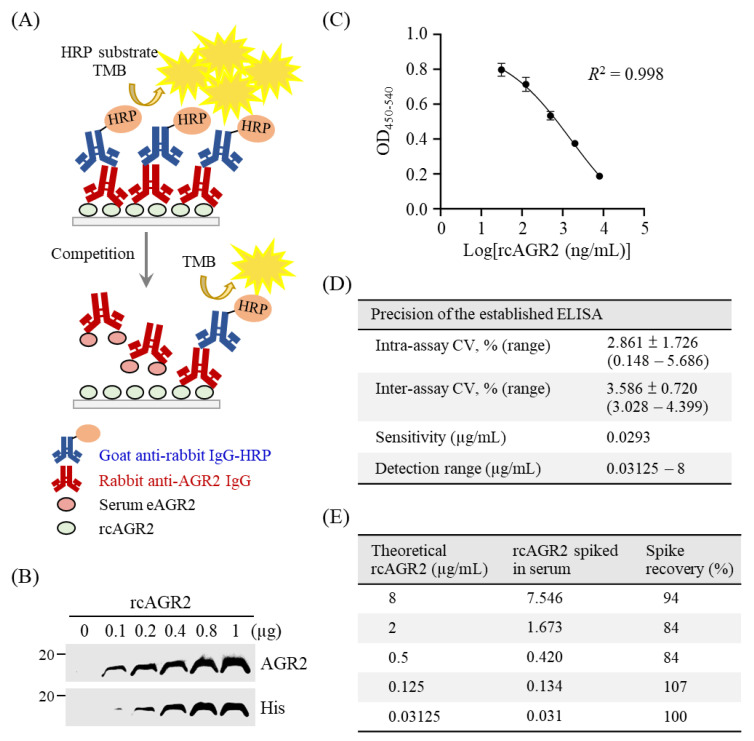
Establishment of a competitive ELISA for detection of serum eAGR2 in MMT dogs. (**A**) Scheme for the established competitive ELISA. (**B**) Identity of recombinant canine AGR2 (rcAGR2) was confirmed by immunoblotting using antibodies specific to AGR2 and 6×His tag, respectively. (**C**) A standard curve with fourfold serially diluted rcAGR2 proteins (8000, 2000, 500, 125, and 31.25 ng/mL) was set for the competitive ELISA. Data are presented as the mean values ± SD across 10 assays. (**D**) Precision of the competitive ELISA. The intra-assay coefficient variant (CV), inter-assay CV, and the detection sensitivity were determined across 10 assays. (**E**) Evaluation of serum influence on eAGR2 detection in the ELISA. A spike recovery test was performed by detecting rcAGR2 spiked at indicated concentrations into canine serum containing undetectable eAGR2. The recovery rate (%) of individual spike rcAGR2 was calculated as indicated.

**Figure 5 animals-11-02923-f005:**
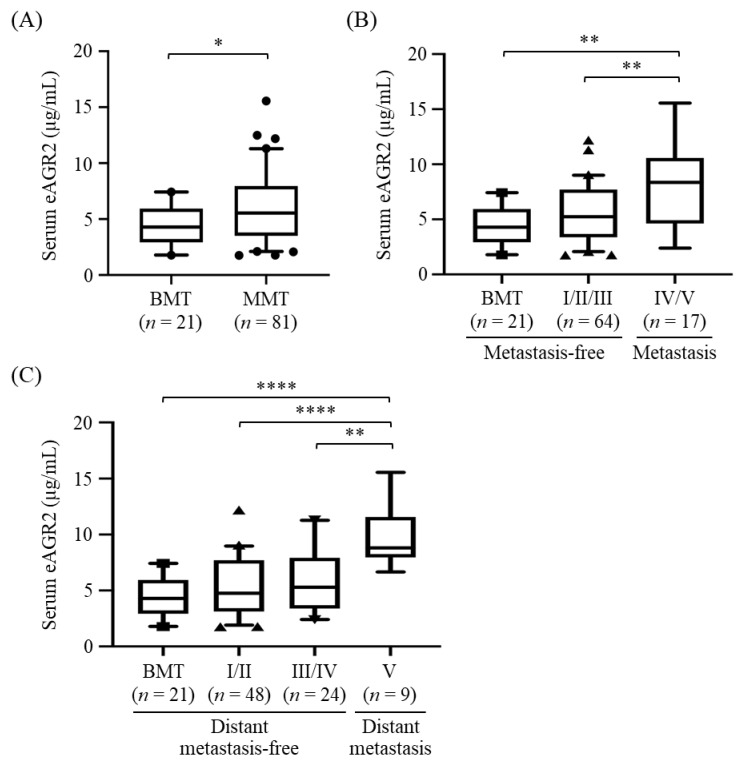
Elevated serum levels of eAGR2 in MMT dogs. Serum eAGR2 concentrations were compared between (**A**) BMT and MMT cases, (**B**) MMT with metastasis (stage IV/V) and metastasis-free cases (stage I to III), and (**C**) MMT with distant metastasis (stage V) and distant metastasis-free cases (stage I to IV). Results are presented in boxplots, wherein the median (bar), the 95th (upper whisker), and the fifth percentile (lower whisker) of the data are shown. Statistical significance between groups was determined by Mann–Whitney *U* test. * indicates *p* < 0.05, ** *p* < 0.01; **** *p* < 0.0001.

**Figure 6 animals-11-02923-f006:**
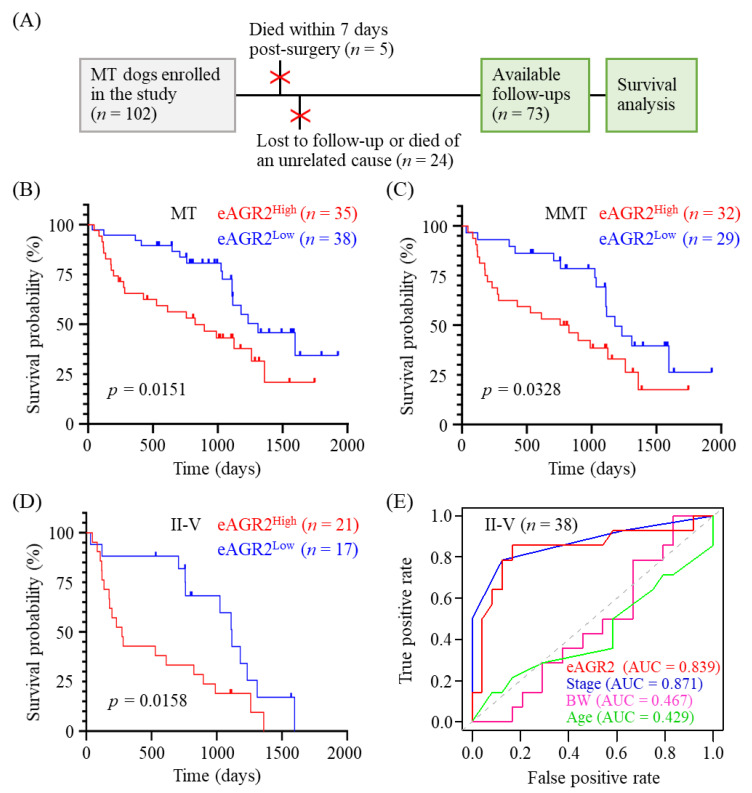
Correlation of serum eAGR2 concentration with adverse MMT prognosis. (**A**) Enrollment of 73 MT dogs with post-surgery follow-ups in the overall survival (OS) analysis. (**B**,**D**) Comparison of OS between MT dogs with low and high concentrations of serum eAGR2. Kaplan–Meier survival plots depict differences in survival probability between eAGR2^Low^ and eAGR2^High^ groups in (**B**) 73 MT cases, (**C**) 61 MMT cases, and (**D**) 38 MMT cases in stage II to V. Statistical significance was determined by the log-rank test. (**E**) Time-dependent receiver operating characteristic curve (ROC) of utilizing serum eAGR2 concentration or other clinicopathological factors for predicting OS of dogs with MMTs in stage II to V. Area under ROC (AUC) was calculated as indicated.

**Table 1 animals-11-02923-t001:** Clinicopathological characteristics of 102 female MT dogs enrolled in serum eAGR2 assessment.

	Benign Mammary Tumor (BMT)			Malignant Mammary Tumor (MMT)		
Group	Case Number	Median (Range) Age, Year	Median (Range) B.W., Kg	Case Number	Median (Range) Age, Year	Median (Range) B.W., Kg
Total	21	8.4 (3–16)	5.4 (1.28–38)	81	10 (2–16)	7.1 (1.7–44.6)
Breed						
Pedigree	19	8 (3–15)	5.4 (1.28–38)	66	10 (2–16)	5.59 (1.7–3.8)
Mixed	2	14.5 (13–16)	18.4 (13.4–23.4)	15	13 (8–16)	15.6 (2.4–44.6)
Neuter						
Yes	6	10 (9–13)	7.2 (2.72–19.4)	34	9 (7–14)	7.1 (2.1–44.6)
No	15	8 (3–16)	5.4 (1.28–38)	47	10 (4–16)	5.9 (1.7–30.4)
MMT subtype						
Simple carcinoma	-	-		45	10.5 (2–16)	7.89 (1.95–44.6)
Complex carcinoma	-	-	-	26	10 (7–16)	7.1 (3.3–29.8)
Multiple MMTs ^1^	-	-		10	9 (7–16)	5 (2.4–30.4)
Clinical stage						
I	-	-		33	9 (2–16)	4.8 (1.7–36.8)
II	-	-		15	11 (8–16)	6.5 (1.95–44.6)
III	-	-		16	12 (8–16)	13 (2.7–29.8)
IV	-	-		8	12.5 (8–15)	11.9 (4.2–38)
V	-	-		9	10 (3.5–14)	12.9 (3.8–26.4)

^1^ The case with more than one type of MMT, including simple carcinoma, complex carcinoma, malignant myoepithelioma, fibrosarcoma, and spindle cell carcinoma.

**Table 2 animals-11-02923-t002:** Correlation between patient signalment and serum eAGR2 concentration.

Characteristics	Case Number	eAGR2 (µg/mL) ^1^	*p*-Value
Age (years) ^2^			
BMT			
<10	12	4.34 ± 1.934	0.831
≥10	9	4.54 ± 1.879	-
MMT			
<10	33	5.58 ± 2.788	0.419
≥10	48	6.23 ± 2.957	-
Total			
<10	45	4.91 ± 2.625	0.199
≥10	57	5.32 ± 2.869	-
Body weight (kg) ^2^			
<6.7	51	4.91 ± 2.265	0.103
≥6.7	51	5.43 ± 3.162	-
Neuter status ^2^			
No	62	5.11 ± 3.021	0.739
Yes	40	5.24 ± 2.370	-
MMT subtype ^3^			
Simple carcinoma	45	6.32 ± 2.878	0.168
Complex carcinoma	26	5.86 ± 3.072	-
Multiple MMTs	10	4.66 ± 2.224	-
Metastasis ^2^			
No	85	5.20 ± 2.324	0.002 **
Yes	17	7.93 ± 3.682	-
Stage ^3^			
I/II	48	4.76 ± 2.459	0.0007 ***
III/IV	24	5.29 ± 2.734	-
V	9	8.81 ± 2.762	-

^1^ Data are shown as mean ± SD. *p*-value < 0.05 is considered statistically significant; ** *p* < 0.01; *** *p* < 0.001. ^2^ Comparison between groups was determined by Mann–Whitney *U* test. ^3^ Comparison between groups was determined by Kruskal–Wallis *H* test.

## Data Availability

The datasets used and/or analyzed during the current study are available from the corresponding author on reasonable request.

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
