# Peer review of "Serum Level of Tumor-Overexpressed AGR2 Is Significantly Associated with Unfavorable Prognosis of Canine Malignant Mammary Tumors"

_animals, 2021, doi:10.3390/ani11102923_

Round 1

Reviewer 1 Report

This is an interesting paper on a novel topic, the relationship of AGR2 expression to prognosis for mammary cancer in dogs. As I understand it, the goal was to determine whether eAGR2 expression might serve as an independent prognostic marker for dogs with malignant mammary tumours. The paper would benefit from some editorial attention to English, although it is written well overall. The conclusion is overstated “The present study also sheds new lights on the progression of canine MMT with regard to AGR2” as association has been identified but not cause and effect, and the levels of AGR2 overlap considerably between BMT and MMT sera. With some self-critique by the authors on flaws in the study, I think the paper is worth publishing. I have a number of specific comments or questions below. Section 2 – not clear from the text – was this a prospectively planned study? Section 2.1 – what sort of surgical resection was done with the cases? Did the cases have additional treatment that might have influenced survival e.g. chemotherapy, further surgery? Please clarify. Section 2.2 • First 3 sentences belong in Results section - (“A total of 102 female dogs ……. in Table 1.”) • Table 1 belongs in Results section. • Please indicate what clinical staging scheme was used – if identical to a published scheme, then please provide the reference. • Was the serum collected from the dogs before or after surgery? • For metastatic tumours do you mean metastatic at diagnosis? Or metastases discovered later? Section 2.3 Please explain a little more about immunohistochemistry – how did you select which sections of tumour to evaluate? Did the evaluation include several fields of view, and at what magnification? What was the source of the control tissues? Section 2.10 Is “expense media” is a tissue culture term unfamiliar to me – possibly a confusion in translation? I would use “spent media”, or “conditioned media”. Figure 1 • For IHC, how were the 34 mammary tumour cases selected from the 81 malignant cases collected? • Were the control cases and tumour cases age-matched? • Did the authors analyse benign tumours with IHC? • In (A) which image is of aberrant AGR2 expression? • Could the authors please indicate which part of the 100x images is displayed in the 400x images. The 400x images should be derived from the 100x field. Figure 2 • please clarify – where immunoblot of normal, benign tumour and malignant tumour are grouped, does this mean that all three tissue types were from the one case? • Please clarify Stage in the figure legend – is it necessary to specify stage in this figure? If not, then remove it. • Why was AGR2 tissue expression measured by immunoblot in only 11 animals? How were these animals chosen? Section 2.11 – probably need to add Kruskall Wallis H test for evaluation of groups …. Figure 3A – the immunostaining for CF41.Mg appears to be either nuclear or diffusely cytoplasmic, not, as in line 274 “...intracellular compartment resembling the ER localization”. The figure legend is more accurate in simply stating AGR2 expression is intracellular in these cell lines. Section 3.2 • Line 277 – “culture” not “cultured” • line 278 – conditioned media is also known as spent media – delete “expense”. For the rest of the paper please use “conditioned media” or “spent media”, not “expense media” Figure 4 B – what samples were used for this figure? Lines 406 /7– replace “reveals” with “is the first to reveal” and delete “for the first time”. For the survival time analyses please could you state the median and range. I assume that the survival curves included all-cause deaths, and have no problem with that. But do you know how many of the cases died or were euthanased as a result of their MMT?

Author Response

Please see the authors' responses in the attached file.

Reviewer 2 Report

This manuscript describes that serum eAGR2 levels are significantly correlated with malignant mammary tumours (MMT) progression and remote tumour metastasis. The authors indicate that serum eAGR2 level is significantly associated with an adverse outcome of MMT dogs and holds a predictive potential in MMT prognosis. Overall, this article is well-written. This work still requires additional experimental validations to increase the robustness of the data.

Major concerns

1 - In Figure 3A, the authors demonstrate that AGR2 was primarily detected at the intracellular compartment, resembling the ER localization. More information on the ER localization must be included. For example, with specific compartment markers (ER, cis-Golgi, ....).

Minor concerns

1 - The figures 1B and D are not described in Results part.

2 - Likewise, the figures 2A, 2B and 2C are not described in Results part.

Author Response

(The authors gave the same response as above.)
